# POST-TRAINING SPARSE ATTENTION WITH DOUBLE SPARSITY

## ABSTRACT

Long-context inference of Large Language Models (LLMs) is known to be challenging due to the excessive Key-Value(KV) cache accesses. This paper introduces "Double Sparsity," a novel post-training sparse attention technique designed to alleviate this bottleneck by reducing KV cache access. Double Sparsity combines token sparsity, which focuses on using only the important tokens for computing self-attention, with channel sparsity, an approach that uses important feature channels for identifying important tokens. Our key insight is that the pattern of channel sparsity is highly static, allowing us to use offline calibration to make it efficient at runtime, thereby enabling accurate and efficient identification of important tokens. Moreover, this method can be combined with offloading to achieve significant memory usage reduction. Experimental results demonstrate that Double Sparsity can achieve $\frac{1}{16}$ sparsity with minimal impact on accuracy across various tasks with different architectures including MHA, GQA, MoE and vision language model. It brings up to a $14.1\times$ acceleration in attention operations and a $1.9\times$ improvement in end-to-end inference on GPUs with various batch sizes. With CPU offloading under extremely long-context settings (e.g., 256K), it achieves a decoding speed acceleration of $16.3\times$ compared to state-of-the-art solutions. Our code is integrated into a widely-used framework SGLang and deployed in real-world workloads.

## 1 INTRODUCTION

Large Language Models (LLMs) have significantly advanced machine learning capabilities, enabling a wide range of applications from natural language processing to complex problem-solving tasks (OpenAI, 2023; Touvron et al., 2023; Google, 2023). Recent progress has dramatically extended the context window of LLMs from 2K to 1M, enabling more long-context applications (Dubey et al., 2024). However, long-context inference is costly and slow due to its token-by-token generation scheme. This auto-regressive process requires loading the entire previous Key-Value (KV) cache to generate the next token, making the self-attention layers extremely memory-intensive and time-consuming (Williams et al., 2009; Liu et al., 2024b). Furthermore, simultaneously serving multiple long-context requests within a single batch greatly increases the size of KV cache, making the inference more memory-intensive (Zhao et al., 2024). For example, serving 16 requests of 8k length with a Llama-2-7B model, a 64 GB KV Cache requires 44 ms, which accounts for more than 80% of the total serving time.

In this paper, we explore methods to reduce access to the KV cache during inference, thereby making attention computation more bandwidth-efficient and accelerating its execution. Our focus is on post-training methods that can be directly applied to a pre-trained model to provide wall-clock acceleration without requiring excessive additional training or fine-tuning overhead. Prior work has attempted to leverage quantization (Hooper et al., 2024; Liu et al., 2024b), compression (Nawrot et al., 2024), and sparsity (Zhang et al., 2024; Anagnostidis et al., 2024; Ge et al., 2024; Ribar et al., 2023) to achieve these goals. Among them, sparsity holds significant potential if a high sparsity ratio can be achieved. The intuition of sparsification is that not every token is equally important for decoding the next token. Therefore, during the decoding process, we can rely on a small subset of important tokens to compute the self-attention, achieving nearly the same results. While the approach of sparse attention seems intuitive, previous research has struggled to find a post-training sparse attention method that maintains high accuracy while being runtime-efficient.

The primary challenge in post-training sparse attention lies in accurately and efficiently identifying important tokens. A naive approach entails calculating the entire attention weight matrices and then sorting the tokens based on the accumulated attention weights. Although this method can precisely identify important tokens, it fails to offer a runtime speedup, as it requires computing the full attention weight matrices, which is precisely the step we aim to avoid. Previous studies have proposed various methods for selecting important tokens; however, these methods either lead to significant accuracy losses or fail to achieve practical wall-clock acceleration. Notably, H2O (Zhang et al., 2024) employs a dynamic strategy that maintains a small, fixed-size cache of important tokens during the decoding steps. Due to its limited size, it must evict tokens during these steps, and once evicted, these tokens cannot be reinstated if they become important later. This approach can lead to significant accuracy degradation since it cannot predict which tokens will be important in the future. Quest (Tang et al., 2024) and SparQ (Ribar et al., 2023), in contrast, retain all tokens and dynamically select important ones at each decoding step, with different importance estimation methods. Quest proposes a page-based estimation method, which is efficient but fails to accurately identify important tokens. SparQ, while being more accurate, falls short of achieving the desired speedup and incurs considerable memory overhead. Therefore, designing a both efficient and accurate estimation method remains a significant challenge.

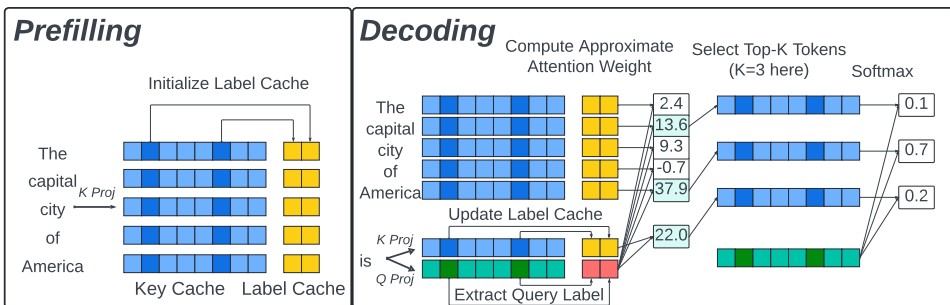

Figure 1: An overview of the Double Sparsity algorithm, including label cache initialization during the prefill stage and sparse attention during the decode stage. Please refer to Section 4 for details.

We propose "Double Sparsity," a method that leverages both token sparsity and channel sparsity to achieve accurate and efficient post-training sparse attention. Token sparsity refers to the sparse attention method mentioned above (Zhang et al., 2024), which uses only important tokens to compute self-attention. Channel sparsity estimates the importance of tokens at runtime using heavy channels in embedding. Our key insight is that while token sparsity is highly dynamic, channel sparsity exhibits static patterns, enabling us to identify and select important channels through offline calibration. This static channel sparsity thus provides an efficient way to achieve dynamic token sparsity at runtime. Furthermore, once we can quickly identify important tokens for the current layer, we extend this process by predicting the important tokens of the next layer. We achieve this by utilizing the embedding similarity between adjacent layers. This approach enables us to offload the entire KV cache to host memory and prefetch only the important tokens to GPU memory, significantly reducing GPU memory footprint.

We demonstrate that "Double Sparsity" can achieve an $\frac{1}{16}$ token sparsity and an $\frac{1}{16}$ channel sparsity simultaneously while incurring only a negligible accuracy loss across a broad array of benchmarks, including language modeling, question answering, and retrieval tasks. The sparsity directly leads to the reduction of memory access and runtime speedup. "Double Sparsity" accelerates the attention operation by up to $14.1\times$ at a sparsity level of $\frac{1}{16}$ on NVIDIA A10G and A100 GPUs, closely approaching the theoretical acceleration upper bound. It accelerates end-to-end inference in our evaluated workloads by up to $1.9\times$. With offloading and prefetch, it achieves a decoding throughput that is $16.3\times$ higher than the state-of-the-art offloading-based solutions at a sequence length of 256K.

## 2 BACKGROUND

### 2.1 PRELIMINARIES ON SELF-ATTENTION AND NOTATIONS

Attention computation is one of the major bottlenecks in LLM Inference, especially when the sequence length is large (Tay et al., 2022). This is caused by its quadratic computational complexity.

Let $d_h$ denote the head dimension, and $S$ denote the number of tokens. We use the decoding step as an example to illustrate the self-attention computation. Each token carries three tensors to embed its information, which are called query, key, and value. In an attention layer, let $q \in \mathbb{R}^{d_h}$ represents the query tensor for input token, $K \in \mathbb{R}^{S \times d_h}$ represents the key tensor for all tokens, and $V \in \mathbb{R}^{S \times d_h}$ represents the value tensor for all tokens. The attention is obtained through the formula shown below:

$$y = softmax\left(\frac{q \cdot K^T}{\sqrt{d_h}}\right) \cdot V$$

## 2.2 Post-training Sparse Attention

In this work, we introduce the term "post-training sparse attention," analogous to "post-training quantization." Post-training sparse attention refers to techniques that exploit inherent model sparsity, such as token-level sparsity, to accelerate attention calculations without requiring additional training. In the field of LLMs, many works have utilized post-training sparse attention, including H2O, StreamingLLM (Xiao et al., 2024), SparQ and Quest. However, these methods come with significant limitations on either maintaining good accuracy or achieving fast inference (details see Section 3), presenting serious challenges for post-training sparse attention.

## 3 Challenges in Post-Training Sparse Attention

In this section, we discuss prior research on post-training sparse attention, identifying the challenges and shortcomings that have prevented these approaches from achieving their full potential. More related works are included in Section 7.

### 3.1 Retrieval Accuracy

One of the most challenging issues for post-training sparse attention is maintaining retrieval accuracy. For instance, StreamingLLM discards earlier tokens except for a few tokens at the beginning, which are called attention sinks, while H2O selectively drops tokens based on previous attention scores. Although discarding tokens can accelerate computations, this exclusion leads to the loss of critical information, potentially compromising the model's retrieval accuracy. As highlighted in Jelassi et al. (2024), this issue is inherent to techniques that rely on discarding tokens, prompting the exploration of sparse attention methods that preserve access to the complete KV cache.

### 3.2 Bandwidth Friendliness

Achieving wall-clock speedup poses a greater challenge while maintaining model retrieval accuracy, particularly because some post-training sparse attention techniques are not hardware-friendly. For instance, SparQ retains the complete KV cache and computes attention selectively on a subset of the KV cache based on the estimation with query. This approach theoretically allows for acceleration while maintaining accuracy. However, SparQ's estimation method uses heavy channels to approximate attention scores, which causes non-contiguous memory access at the granularity of 2B for FP16. Such irregular access can substantially waste memory bandwidth, as GPU is designed to efficiently deal with continuous 16B memory loading (NVIDIA, 2020). As a result, despite being designed to accelerate processing, SparQ achieves only a modest $1.3 \times$ speed increase in attention computations. We further quantify this effect in Appendix B. Therefore, it is crucial to develop an algorithm that ensures continuous memory access patterns to accelerate attention while preserving accuracy.

### 3.3 Memory Footprint

Besides the efficiency of memory bandwidth, long-context inference is still challenging because of the substantial memory capacity required to store the KV cache, as post-training sparse attention can't prune tokens to assure great accuracy. E.g., with Llama2-7B and a context length of 128K, 64GB is still needed even with 1/16 token sparsity. To mitigate this large memory footprint, FlexGen (Sheng et al., 2023b) offloads the entire KV cache on the CPU and loads it to the GPU layer-by-layer during the decoding process. However, the communication overhead of loading the KV cache for all previous

tokens within a single layer is still large enough to affect overall inference efficiency. In sparse attention, considering that important tokens constitute just a small fraction of all tokens, the time taken to load these specific tokens to the GPU is considerably less than the time required for loading the entire KV cache. By accurately predicting the important tokens and efficiently managing when and how data is transferred and processed, it's possible to significantly reduce both the time and memory overhead typically associated with maintaining a full KV cache.

To address these challenges, we propose two post-training sparse attention techniques. In Section 4, we introduce Double Sparsity, which accelerates attention by up to $16 \times$ with minimal additional memory consumption. In Section 5, we present Double Sparsity-Offload, which reduces memory usage to $\frac{1}{16}$ of the original without increasing latency.

# 4 DOUBLE SPARSITY

Based on the insights of Section 3, we propose Double Sparsity, a hardware-friendly and bandwidth-efficient post-training sparse attention mechanism. This approach overcomes the challenges highlighted in previous post-training sparse attention techniques by ensuring no loss of information, as it maintains the **entire KV cache**. To avoid the cache misses associated with runtime sorting, Double Sparsity utilizes **offline calibration** to pre-determine outlier channels for each transformer layer. A compact **label cache** is employed to store outlier channel values from the Key cache, optimizing memory access patterns to leverage GPU's preference for contiguous memory access. Algorithm 1 and Figure 1 illustrate the decoding process of Double Sparsity.

---

**Algorithm 1** Double Sparsity Decode

**Input:** Query vector $q \in \mathbb{R}^{d_h}$, Key cache $K \in \mathbb{R}^{S \times d_h}$, Value cache $V \in \mathbb{R}^{S \times d_h}$, Label cache $L \in \mathbb{R}^{S \times r}$, Calibrated channel index list $C \in \mathbb{N}^r$, Top-k token number $k$
**Output:** Sparse attention output $y$

1: $q_{\text{label}} \leftarrow q_{[C]}$
2: $L_{[-1,:]} \leftarrow K_{[-1,C]}$
3: $\hat{s} \leftarrow q_{\text{label}} \cdot L$
4: $i \leftarrow \text{argtopk}(\hat{s}, k)$
5: $s \leftarrow \text{softmax}\left(\frac{q \cdot K^T[i,:]}{\sqrt{d_h}}\right)$
6: $y \leftarrow s \cdot V[i,:]$
7: **return** $y$

---

## 4.1 OFFLINE CALIBRATION

Offline calibration is a commonly used technique to identify channel sparsity, particularly effective for pinpointing outlier channels. For example, AWQ (Lin et al., 2023) utilizes offline calibration to identify salient weight channels that significantly impact model performance. Inspired by this approach, we employ offline calibration to pre-determine the channels that most influence attention scores. Attention computation can be expressed as $A = Q \cdot K^T$, which can be broken down into $A = \sum_i^{d_h} S_i$ where $S_i = Q_i * K_i$. Due to channel sparsity, only a few $S_i$ have a significant impact on $A$. Therefore, by conducting offline calibration on a small validation set, we can efficiently identify these critical channels by computing the $\arg\max_i S_i$. Figure 7a in Appendix A illustrates the outlier channels identified by AWQ and Double Sparsity.

To validate the efficacy of outlier channels identified through offline calibration, we conducted a comparison in Appendix A between the outlier channel indices derived from offline calibration and those determined during the online decoding process. A significant overlap between the two sets underscores the reliability of offline-calibrated outliers. Figure 7b illustrates this relationship. An observation from the comparison is that when the ratio surpasses 0.25, the overlap reaches 0.95.

## 4.2 FORWARDING WITH LABEL CACHE

After identifying the outlier channel indices, it becomes crucial to access them efficiently. Reading these channels directly from the Key cache can lead to non-contiguous memory accesses, which significantly under-utilized the bandwidth. To alleviate non-contiguous memory accesses, we leverage a label cache to store pre-determined heavy channel values. This label cache allows for continuous memory access when computing approximate attention, avoiding the need to retrieve non-contiguous

segments from the Key cache. During the prefilling stage, all heavy channel values from the Key cache are stored in the label cache; in the decoding phase, only the heavy channel values of new tokens are added. Since approximate attention is not sensitive to precision, we can store the label cache in 4-bit. This approach enables us to maintain a label cache that is only $\frac{1}{16}$ the size of the K cache, facilitating contiguous memory access and significantly improving the hit rate of L1/L2 caches, thereby optimizing inference speed and efficiency. In Appendix B, an ablation study was conducted to evaluate the impact of label caches. The results demonstrated that a label cache accelerates decoding speeds by 2 to 4 times compared to configurations without a label cache.

## 5 REDUCING GPU MEMORY USAGE WITH DOUBLE SPARSITY-OFFLOAD

Building upon Double Sparsity, we further propose the Double Sparsity-Offload technique to reduce the large GPU memory footprint introduced by long-context inference. This approach significantly diminishes the memory capacity requirement to $\frac{1}{16}$ of the original KV caches. By optimizing memory footprint, Double Sparsity-Offload enables efficient decoding under extremely long-context settings with limited GPU memory resources.

### 5.1 PREFETCHING TOKENS WITH DOUBLE BUFFER

The Double Sparsity-Offload algorithm introduces a prefetching technique with a double buffer during for decoding process, following the design principles of prior works (Lee et al., 2024; Shi et al., 2024). Our key idea is to utilize inter-layer similarity to predict and prefetch the heavy tokens for the next layer and overlap the computation with memory transfer. The complete KV cache is offloaded to the CPU, while the GPU maintains only the label cache and a double buffer. During the decoding process, each layer processes its embeddings through the next layer's query projection to generate an approximate query for the subsequent layer. This approximate query is then used to compute the next layer's approximate attention. While the current layer's attention and feed-forward network computations are being performed, the tokens corresponding to the approximate attention results for the next layer are loaded to the GPU. This use of double buffering allows for a smooth and efficient overlap of computation and memory transfer.

### 5.2 EMPIRICAL ANALYSIS: EMBEDDING SIMILARITY BETWEEN LAYERS

The feasibility of the Double Sparsity-Offload algorithm is based on the high degree of similarity between embeddings across consecutive layers. To empirically validate this assumption, we conducted an analysis using the Pile validation dataset, applied to the Llama-2-7B model. We measured the cosine similarity of embeddings between every two consecutive layers throughout the model. The results show that apart from the first two layers, the second and third layers, and the very last layers (30 and 31), all other layer pairs exhibited a cosine similarity exceeding 90%, with the majority of layers showing similarities above 95%, as illustrated in Figure 2. These high similarity scores support the viability of utilizing prior layer embeddings to predict queries for subsequent layers in Double Sparsity-Offload.

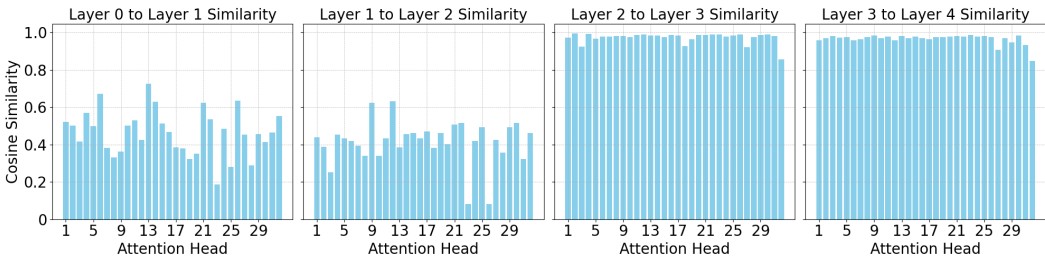

Figure 2: Average cosine similarity of embeddings across all attention heads between layers 0-1, 1-2, 2-3, and 3-4 on the Pile dataset for Llama-2-7B model. The cosine similarity pattern for other consecutive layers except the very last layers is similar to the pattern for layers 2-3 and 3-4.

Table 1: Comparison of sparsity-related techniques. 'SparQ (1xK)' denotes single-dimension storage of the Key cache, while 'SparQ (2xK)' refers to dual-dimension storage of the Key cache. Quest stores max and min representations per page with a page size of 16, indicating $\alpha$ is 1/8.

| Method | On-device Cache Size | Cache IO-Complexity | Min $\beta$ | Speedup |
|---|---|---|---|---|
| H2O | $S \times \beta$ | $S \times \beta$ | 1/5 | Yes |
| SparQ (1xK) | $S$ | $S \times \beta$ | 1/8 | No |
| SparQ (2xK) | $S \times 1.5$ | $S \times \beta$ | 1/8 | Yes |
| AWQ | $S$ | $S$ | 1 | Yes |
| Quest | $S \times (1 + \frac{\alpha}{2})$ | $S \times \beta$ | 1/16 | Yes |
| Double Sparsity | $S \times (1 + \frac{\alpha}{2})$ | $S \times \beta$ | 1/16 | Yes |
| Double Sparsity-Offload | $S \times \frac{\alpha}{2}$ | $S \times \beta$ | 1/16 | Yes |

## 5.3 COMPLEXITY ANALYSIS

To understand the potential speedup of Double Sparsity, we need to analyze its Cache IO-Complexity since attention mechanisms are bandwidth-bounded. Most sparse attention techniques can be simplified into two steps: calculating approximate attention and computing attention over $k$ tokens. We denote $\alpha$ as the ratio of memory access in the first step relative to the K cache size, and $\beta$ as the ratio of memory access in the second step relative to the KV cache size. Memory-wise, the total access of Double Sparsity comprises $O(d)$ bytes for $Q$, $O(S \times r)$ for the label cache, $O(2 \times k \times d)$ for the KV cache, leading to a total of $O(S \times r + 2 \times k \times d) = O(\alpha \times S \times d + 2 \times \beta \times S \times d)$. Given that the approximate attention phase of Double Sparsity does not involve softmax operations, it allows for high parallelism compared to the following step. Therefore, the overall IO complexity of Double Sparsity primarily depends on the latter step, which can be approximated as $O(2 \times \beta \times S \times d)$. This analysis reveals that Double Sparsity's time complexity is linearly dependent on $\beta$, and the extra memory overhead is linearly proportional to $\alpha$. Table 1 summarizes all the sparsity works discussed, specifying their overhead, complexity, and speedup.

## 6 EXPERIMENT

In Section 6.1, we demonstrate that both Double Sparsity and Double Sparsity-Offload maintain robust performance with a sparsity setting of 1/16 across various benchmarks, including Wiki-2 perplexity (Merity et al., 2016), MultifieldQA (Bai et al., 2023), GovReport (Huang et al., 2021), HotpotQA (Yang et al., 2018), TriviaQA (Joshi et al., 2017), NarrativeQA (Kočiský et al., 2017), and Qasper (Dasigi et al., 2021). In key-value retrieval tasks (Li et al., 2023), Double Sparsity significantly outperforms other post-training sparse attention techniques. In Section 6.2, we compare Double Sparsity against state-of-the-art attention and end-to-end implementations. Results show that Double Sparsity achieves up to a 16-fold acceleration in attention mechanisms and up to a twofold increase in overall end-to-end processing speed. Additionally, Double Sparsity-Offload achieves a 16-fold acceleration compared to FlexGen Offload.

### 6.1 ACCURACY EVALUATION

#### 6.1.1 WIKI-2 PERPLEXITY

Wiki-2 perplexity is a benchmark derived from Wikipedia articles, offering a comprehensive test of language modeling. We evaluate Double Sparsity on Wiki-2 with different sparsity levels under different models. Note that a lower perplexity indicates better model performance. Table 2 illustrates the changes in perplexity across different sparsity levels for each model.

To demonstrate the model's performance at different sparsity levels and justify our selection of a sparsity level of 1/16, we constructed a 3D bar chart. According to Figure 8 in Appendix C, a noticeable shift in perplexity is observed as the sparsity level goes beyond 1/16.

To validate the robustness of Double Sparsity, we conducted a series of ablation studies across various model configurations and conditions. Table 3 demonstrates the effectiveness of Double Sparsity across various model sizes, attention mechanisms, and MoE configurations. The sparsity level of

Table 2: Perplexity of models at various sparsity levels. Note the minimal changes in perplexity from sparsity levels 1 to 1/16, with a significant performance gap emerging between levels 1/16 and 1/32.

| Model | Sparsity Level | | | | | |
|---|---|---|---|---|---|---|
| | 1 | 1/2 | 1/4 | 1/8 | 1/16 | 1/32 |
| Llama-7B | 5.68 | 5.69 | 5.69 | 5.72 | 5.80 | 7.66 |
| Llama-2-7B | 5.47 | 5.48 | 5.53 | 5.56 | 5.76 | 12.01 |
| Llama-2-7B (offloading) | 5.47 | 5.48 | 5.54 | 5.57 | 5.86 | 15.29 |
| Llama-3.1-8B | 6.24 | 6.24 | 6.25 | 6.27 | 6.35 | 8.56 |
| Llama-3-70B-Instruct | 5.31 | 5.29 | 5.33 | 5.35 | 5.54 | 13.47 |
| Mistral-7B | 5.25 | 5.25 | 5.26 | 5.27 | 5.37 | 14.55 |

Table 3: Ablation study on different architectural models with different outlier types at 1/16 sparsity level. GQA models are incompatible with K outlier channel. 'random' denotes using random channel.

| Model | Architecture | Original | Double Sparsity | | | |
|---|---|---|---|---|---|---|
| | | | random | q outlier | k outlier | qk outlier |
| Llama-2-7B | Single/MHA | 5.47 | 8.62 | 6.45 | 6.61 | 5.76 |
| Llama-2-7B-chat | Single/MHA | 6.94 | 10.1 | 7.8 | 9.44 | 7.14 |
| Mistral-7B | Single/GQA | 5.25 | 6.06 | 5.79 | N/A | 5.37 |
| Llama-2-70B | Single/GQA | 3.32 | 5.15 | 3.69 | N/A | 5.17 |
| Mixtral-8x7B | MoE/GQA | 3.84 | N/A | 3.84 | N/A | 17.3 |
| Llama-3.2-11B-Vision | VLM | 7.23 | N/A | 8.85 | N/A | N/A |

Double Sparsity can be adaptively specified according to the task. Based on the results of perplexity evaluations, we can designate 1/16 as the common sparsity level.

### 6.1.2 Long Context Benchmarks

We used Llama-3.1-8B-Instruct to evaluate the performance of Double Sparsity across multiple long context benchmarks at various levels of sparsity, comparing its effectiveness with that of StreamingLLM, H2O and Quest. As illustrated in Figure 3, Double Sparsity maintains its performance with nearly no drop in accuracy at a sparsity level of 1/16, outperforming other techniques. We also evaluated Double Sparsity on Llama-2-7B-chat in Appendix D.

### 6.1.3 Key-Value Retrieval

The key-value retrieval benchmark is designed to assess a model's in-context retrieval capabilities. Our experiments compared Double Sparsity against other post-training sparsity techniques, including H2O, StreamingLLM, and Quest. We also tested the performance of Double Sparsity with the Llama-3.1-8B-128k model to observe how accuracy changes as context length increases. As shown in Figure 4, we demonstrate that Double Sparsity significantly surpasses the other techniques in key-value retrieval tasks. Notably, Double Sparsity and Double Sparsity-Offload show equivalent performance, highlighting that the offloading mechanism exhibits almost no decay.

## 6.2 Speedup Evaluation

### 6.2.1 Setups

**Hardware.** Our experiments were conducted on two types of GPUs: the A10G and the A100-SXM.

**Implementation.** For the Double Sparsity Attention, we utilized PyTorch to compute approximate attention and select the top-k tokens. The kernel for attention over top-k tokens was designed using OpenAI Triton. For end-to-end testing, we replaced the full attention mechanism in gpt-fast (PyTorch, 2023b) with our Double Sparsity Attention. For Double Sparsity-Offload, we implemented asynchronous CPU to GPU memory copying using CUDA streams and DGL (Wang et al., 2019)'s gathering kernel.

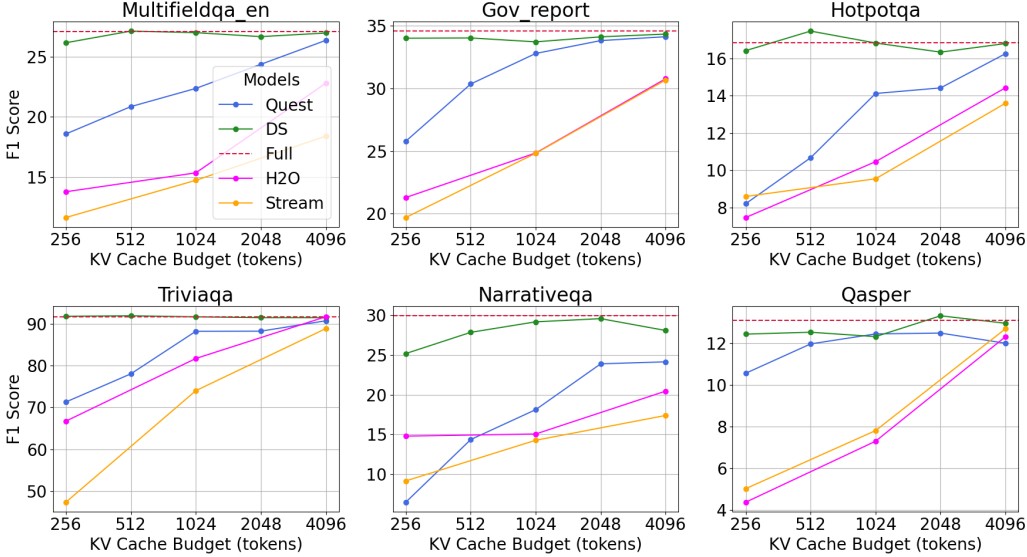

Figure 3: We tested Double Sparsity and other baselines on six long-context benchmarks. 'DS' denotes Double Sparsity, 'Stream' denotes StreamingLLM, and 'Full' indicates the score of normal inference. It is noted that Double Sparsity outperformed all other baselines.

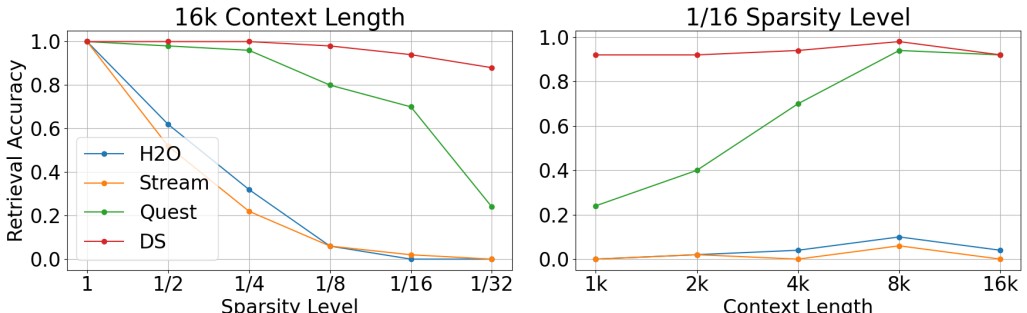

Figure 4: Performance of Double Sparsity and other baselines tested via key-value retrieval, examining different sparsity levels at a fixed 16k context length and various context lengths at a fixed 1/16 sparsity level. Double Sparsity outperformed all other baselines in both settings.

**Workload.** We selected Llama-2-7B as base model and focused on high-workload scenarios to push the limits of Double Sparsity. This included a range of batch sizes from 4 to 32 and sequence lengths from 1024 to 16384. For Double Sparsity-Offload, we extended testing to extreme conditions on the A100 GPU, exploring sequence lengths from 64K to 256K. Given that gpt-fast's KV cache is pre-allocated, the tokens-per-second throughput depends solely on the batch size and sequence length.

**Baseline.** For attention acceleration evaluations, we use the 'scaled_dot_product_attention' as our baseline. This implementation ranks among the fastest attention mechanisms, dynamically allocating computation among the most efficient options including FlashAttention-2 (Dao, 2023), Memory-Efficient Attention (Lefaudeux et al., 2022), and the top-performing kernels from the PyTorch team. In the end-to-end speed evaluations of Double Sparsity, gpt-fast (PyTorch, 2023a) serves as the baseline, distinguished as the state-of-the-art for Llama models on the A100 GPU. It offers exceptionally low latency and throughput that surpasses that of the huggingface transformers by tenfold. For evaluating Double Sparsity-Offload, we compare it against FlexGen Offloading, which shares the same gpt-fast codebase and memory footprint.

**Other Settings.** Given Double Sparsity's focus on the attention mechanism, both weights and activations were set to FP16 precision. Furthermore, considering the limitations imposed by Triton kernels on Torch compile options, neither Double Sparsity nor gpt-fast employed the Torch compiler.

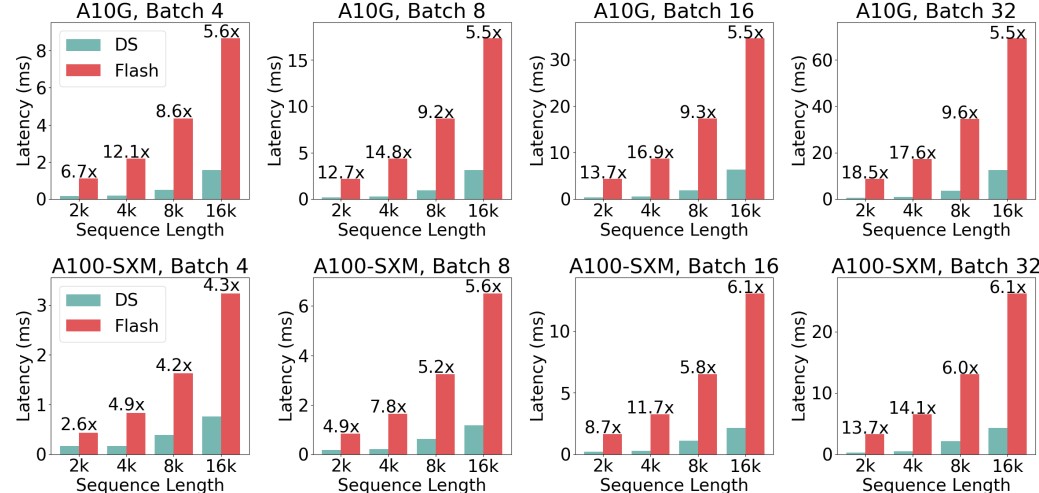

Figure 5: Latency and speedup of Double Sparsity Attention at various batch sizes and sequence lengths. 'DS' indicates double sparsity attention. 'Flash' indicates the 'scaled_dot_product_attention', which is the fastest of FlashAttention-2 and Memory-Efficient Attention.

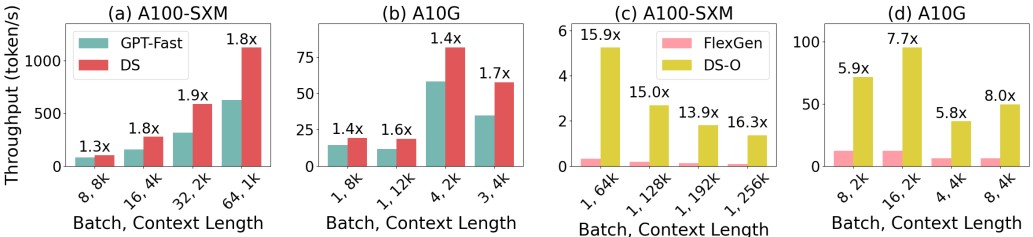

Figure 6: Throughput (token/s) and speedup of Double Sparsity (Offloading) in end-to-end scenarios.

### 6.2.2 ATTENTION OPERATOR SPEEDUP

Figure 5 provides a comprehensive view of the latency and speedup of Double Sparsity compared to 'scaled_dot_product_attention' across different batch sizes and sequence lengths. On the A10G GPU, every case achieves at least a $5\times$ speedup, with more than half exceeding $9\times$. Notably, Double Sparsity achieves a linear speedup at a sequence length of 4096 with large batch sizes. On the A100 GPU, nearly all cases see at least $4\times$ faster processing, with larger batches reaching up to $10\times$ speedup. The greater speedup for smaller batches on the A10G might be due to the launch time of Triton kernels, which becomes significant when the kernel execution time on the A100 is short.

### 6.2.3 END-TO-END INFERENCE SPEEDUP

Figure 6 (a)(b) presents the throughput comparison between Double Sparsity and gpt-fast, measured in tokens per second across various batch sizes and sequence lengths. We deployed the Llama-2-7B model and maximized memory usage to achieve high workload conditions. The results indicate that Double Sparsity yields a minimum speedup of 1.3x across all tested conditions. In certain scenarios, the speedup approached twofold, showcasing Double Sparsity's overall efficiency.

In Figure 6 (c)(d), we compare the throughput of Double Sparsity-Offload to FlexGen under a constrained memory footprint, set at 1/16 of a full KV cache for both methods. Both techniques utilize a double buffer for asynchronous data copying. The results show that Double Sparsity-Offload achieves a 4-8$\times$ speedup over FlexGen under regular workloads, and a 16$\times$ speedup in scenarios with long texts ranging from 64K to 256K in sequence length.

## 7 RELATED WORK

**Sparse Attention Inference**    Due to the significant memory-intensive nature of self-attention, many studies have focused on exploiting sparsity to accelerate the inference process. These efforts can be categorized under three main criteria: 1) static or dynamic sparse patterns; 2) the presence of token eviction; 3) accelerating prefilling or decoding. StreamingLLM (Xiao et al., 2024) and LM-Infinite (Han et al., 2023) utilize static sparse patterns with token eviction to accelerate decoding. These approaches achieve inference acceleration by preserving only a small and fixed number of initial tokens along with local tokens. H2O (Zhang et al., 2024) and Scissorhands (Liu et al., 2024a) employ dynamic sparse patterns with token eviction for decoding, preserving only a small fraction of the KV cache called heavy hitters according to accumulated attention scores, while FastGen (Ge et al., 2024) uses adaptive sparse attention patterns for different attention heads. MInference (Jiang et al., 2024) serves as a prefilling acceleration method that retains all tokens. It first identifies sparse patterns within the model via offline calibration, and then leverages these identified patterns to accelerate the pre-filling stage. SparQ (Ribar et al., 2023) and Quest (Tang et al., 2024) implement dynamic sparse decoding while preserving all tokens without eviction. SparQ filters the important tokens using heavy channels of incoming query and Key cache. Quest segments token into multiple pages and proposes a page-based estimation method to utilize sparsity in self-attention.

**Sparse Attention Training**    There are also many efforts to reduce attention complexity through training (Qiu et al., 2020; Ding et al., 2023; Tay et al., 2020; Chen et al., 2021). For example, Sparse transformer (Child et al., 2019) reduces the complexity to $O(n\sqrt{n})$ by introducing sparse factorization of the attention matrix. Reformer (Kitaev et al., 2019) achieves $O(n \log n)$ complexity via locality-sensitive hashing. Longformer (Beltagy et al., 2020), BigBard (Zaheer et al., 2020), and Linformer (Wang et al., 2020) further reduce the complexity to linear. Linear attention architectures have also been proposed in Katharopoulos et al. (2020).

**Other Attention and Inference Optimizations**    Despite efforts to sparsify the self-attention computation, there are many other optimizations for attention efficiency. Common techniques include quantization and compression (Hooper et al., 2024; Liu et al., 2024b; Kang et al., 2024; Nawrot et al., 2024), efficient attention architecture like multi-query attention (Shazeer, 2019) and group-query attention (Ainslie et al., 2023), and memory-efficient attention algorithms (Rabe & Staats, 2021; Dao et al., 2022). Alternatives to transformers include using the state space model to remove the attention mechanism (Gu et al., 2021). Other common inference optimizations for LLMs include batching Yu et al. (2022), memory optimizations Sheng et al. (2023b); Kwon et al. (2023); Aminabadi et al. (2022), parameter sharing Sheng et al. (2023a); Chen et al. (2023), speculative decoding Stern et al. (2018); Leviathan et al. (2023); Miao et al. (2023), scheduling Han et al. (2022); Agrawal et al. (2023); Patel et al. (2023); Zhong et al. (2024), quantization Xiao et al. (2023); Lin et al. (2023); Dettmers et al. (2022); Frantar et al. (2022), and sparsification Frantar & Alistarh (2023).

## 8 FUTURE DIRECTIONS AND CONCLUSION

**Future Directions.** Double Sparsity can be integrated with other sparse attention techniques. For instance, Quest could harness Double Sparsity using heavy channels instead of max-min representations to reduce total memory access. Similarly, MInference could employ heavy channels in place of pooling when computing block sparse patterns to increase accuracy. Despite the progress made with Double Sparsity, several limitations remain that reveal promising directions for future research. It is challenging to perfectly overlap communication with computation. Enhancing asynchronous capabilities to mask communication overheads presents a promising direction that allows for significant acceleration with a minimal memory footprint.

**Conclusion.** In this work, we introduced Double Sparsity and Double Sparsity-Offload, innovative post-training sparse attention techniques. Double Sparsity leverages offline calibration and label cache to achieve nearly lossless performance across various benchmarks at a 1/16 sparsity level. Performance tests showed that Double Sparsity could accelerate attention computations by up to $16\times$ and achieve an end-to-end speedup of $1.9\times$. Double Sparsity-Offload significantly reduced KV Cache memory usage to 1/16, outperforming the throughput of previous SOTA offloading techniques by 16 times.

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

# A OFFLINE CALIBRATION ILLUSTRATION

The x-axis of the Figure 7b denotes the ratio of the selected top-k channels to the total number of channels, while the y-axis quantifies the degree of overlap between the offline and online outliers.

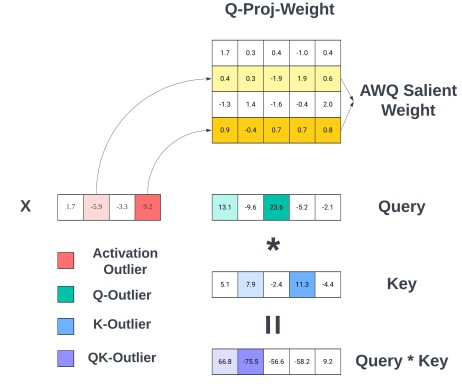

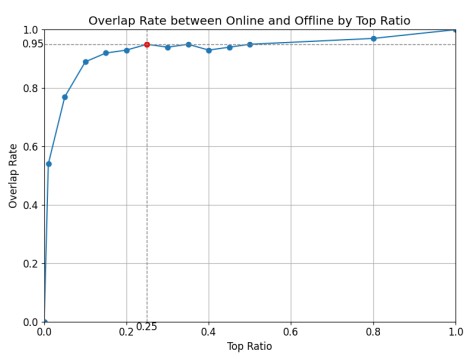

(a) Outlier channels of AWQ and Double Sparsity.

(b) Outlier channel overlap rate between offline calibration and online decoding.

Figure 7: Analysis of Double Sparsity calibration in identifying outlier channels

# B FORWARD WITHOUT LABEL CACHE

To investigate the significance of the label cache in the forward pass of Double Sparsity, we conducted experiments comparing performance with and without the label cache. As depicted in Table 4, label caches significantly enhance the forward pass, yielding a speedup ranging from 2 to 4 times.

Table 4: Latency comparing performance With and Without Label Cache.

| Batch | Seq Len | With Label Cache (ms) | Without Label Cache (ms) | Speedup |
|-------|---------|-----------------------|--------------------------|---------|
| 4     | 2048    | 0.165                 | 0.279                    | 1.7     |
| 4     | 4096    | 0.181                 | 0.559                    | 3.1     |
| 4     | 8192    | 0.504                 | 1.250                    | 2.5     |
| 4     | 16384   | 1.550                 | 3.000                    | 1.9     |
| 32    | 2048    | 0.467                 | 1.960                    | 4.2     |
| 32    | 4096    | 0.983                 | 3.950                    | 4.0     |
| 32    | 8192    | 3.600                 | 9.540                    | 2.6     |
| 32    | 16384   | 12.600                | 24.000                   | 1.9     |

## C  PERPLEXITY SELECTION ILLUSTRATION

Figure 8 uses token-level sparsity as the x-axis, channel-level sparsity as the y-axis, and 10-perplexity values as the z-axis, where higher bars indicate better performance. A sudden shift in perplexity is observed as the sparsity level goes beyond 1/16.

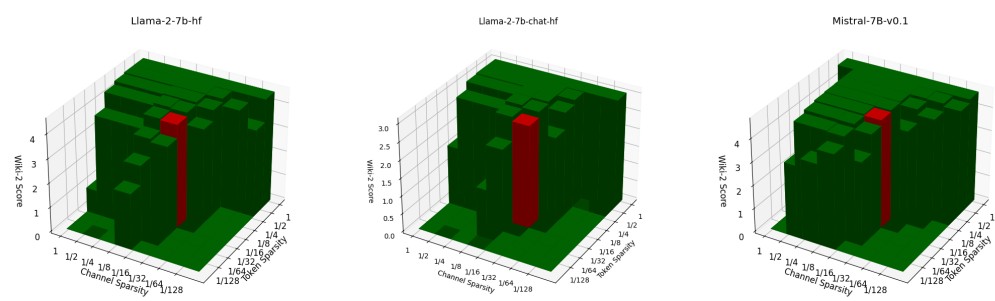

Figure 8: Perplexity of models at different token-sparsity and channel-sparsity levels. Notably, the red bars, representing a sparsity level of 1/16 for both token and channel, show that the model's performance remains largely consistent with the original model at this level.

## D  LLAMA2 LONGBENCH EVALUATION

We also used Llama-2-7B-chat to evaluate the performance of Double Sparsity across multiple long context benchmarks at various levels of sparsity as shown in Figure 9.

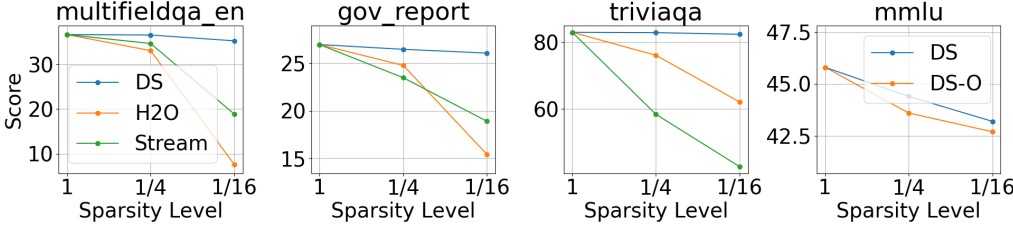

Figure 9: Performance of different techniques across various sparsity levels for Llama-2-7B. 'DS' and 'DS-O' refer to Double Sparsity and Double Sparsity-Offloading. 'Stream' refers to Streaming-LLM.