# OpenReview forum: "Post-Training Sparse Attention with Double Sparsity"
_ICLR.cc/2025/Conference — Submitted to ICLR 2025_

### Official Review · Reviewer_Tc6k · 2024-10-27

**Soundness:** 3
**Presentation:** 3
**Contribution:** 2
**Rating:** 5
**Confidence:** 5

**Summary:**

This work focuses on a issues in Sparse Self-Attention at post-training stage and proposes the double sparsity from both token-level and channel-level based on previous observations. It uses two mechanisms: (i) select important channels through offline calibration in CPU, and (ii) select specific tokens to compute attention score during runtime inference. Experiments show that both accuracy (PPL, long-context benchmark) and efficiency (kernel-level, e2e) outperforms SOTA sparse attention works and dense models.

**Strengths:**

Long-context inference is an important research direction nowadays, where attention occupies a bottleneck for compute/memory. In the calibration stage, this work draws on insights based on AWQ and uses offline pre-processing of outliers in the channel dimension. In the post-training/inference stage, the tokens that really need to be calculated are selected based on the attention scores of the topk. Compared with StreamingLLM, H2O, and Quest, comprehensive and detailed experimental data on Wiki-2 PPL and Longbench show better accuracy, which indeed proves that the work is reliable.

**Weaknesses:**

1. I’m curious about the resources and time required for the offline calibration phase in your preprocessing, specifically for a model like Llama-3.1-8B. Ideally, we would not want this process to consume too many resources or take too much time.

2. Could you elaborate on which open-source experiment the key-value retrieval benchmark is based on? Additionally, could you provide comparison results with the baseline and related works on the latest Needle in a Haystack (https://github.com/gkamradt/LLMTest_NeedleInAHaystack) and RULER (https://github.com/zhang677/RULER)?

3. It seems to me that the concept of channel-sparsity is quite similar to some ideas in AWQ and QServe, while token-sparsity reminds me of previous work like Quest. Could you explain in detail your advantages and improvements in these areas?

**Questions:**

My questions are written under the weaknesses section. If they can be addressed, I'd be happy to raise the score.

---

### Official Review · Reviewer_uWdc · 2024-10-30

**Soundness:** 4
**Presentation:** 2
**Contribution:** 3
**Rating:** 8
**Confidence:** 5

**Summary:**

The paper designs offline calibration and label cache to make token and channel sparsity relatively more hardware friendly compared to existing sparse attention training and inference works.
With such design, the method could achieve decent wall-clock speed-up and memory reduction, while maintaining good context retrieval capabilities.

**Strengths:**

1. The insight into how existing KV management works are not feasible for actual hardware deployment is sharp and valid in its class. This direction was supposed to solve actual production scenarios, but most of the existing work actually overlook the hardware nature, e.g., IO pattern, fused operation, fragmentation, and so forth. It's good to say this paper point out these issues, which could make future work more solid.
2. The proposed solution is effective, as it generates much better wall-clock speed-up and memory saving. More importantly, the method is integrated SGLang. To the best of my knowledge, most of the methods in class can hardly be integrated into actual serving system like vLLM, SGLang, and TensorRT.
3. Performance preserving is plausible.

**Weaknesses:**

1. Writing seems to be rushed and very unclear in many places. Please include a notation table and a taxonomy table, or at least declare the notation before or shortly after usage.
2. The long context evaluation may be done on longer contexts, e.g., 128k, and more challenging tasks, e.g., RULER and BABILONG. From my perspective, as long as the method can work on 128k context with the 1/16 sparsity, I would be rather impressed. More complex eval can be saved for future work.
3. It would be good to see how more comprehensive benchmarks, e.g., MMLU, and long generation tasks, e.g., ChatArena. I'm generally fine with this point as some previous methods in the class already tested it.
4. It would be good if [1] can be cited, as it stated similar hardware insights and published earlier than this one, e.g., it's infeasible to manage small memory chunks at arbitrary location, KV management should be hardware-aware and compatible with actual inference/training frameworks like vLLM/SGLang/Megatron. I assume two works are parallel and it's okay if [1] is not included.

[1] S2-Attention: Hardware-Aware Context Sharding Among Attention Heads

**Questions:**

1. I'm curious if there's a sparsity law, e.g., what sparsity level can be reached at what context length, for what task, and model scale? Also, I would assume larger model might have more to discuss for channel sparsity, both from engineering perspective and from quality preserving perspective.
2. How well does the method work with tensor parallel and pipeline parallel? I'm curious how the speed/memory bonus go with tp and pp size?

---

### Official Review · Reviewer_Qb85 · 2024-11-03

**Soundness:** 2
**Presentation:** 2
**Contribution:** 2
**Rating:** 3
**Confidence:** 4

**Summary:**

This paper introduces Double Sparsity, a technique that jointly prunes tokens and channels by leveraging the static nature of channel sparsity, enabling offline calibration for runtime efficiency and precise identification of key tokens. The authors also propose an efficient implementation by offloading the entire KV cache to host memory and prefetching only essential tokens to GPU memory, significantly reducing GPU memory usage.

**Strengths:**

+ Jointly channelwise and tokenwise pruning
+ Real GPU implementation
+ Good system performance

**Weaknesses:**

- While joint channelwise and tokenwise pruning is novel, it closely resembles headwise and tokenwise pruning, which has been explored in previous work [1], raising concerns about the paper's originality.
- The evaluation lacks comprehensiveness, as several LLM pruning baselines are not included ([2], [3], [4]).
- The writing quality requires substantial improvement, with several inaccuracies in mathematical notation. For instance, in line 200, Qi and Ki are not clearly described—it's unclear if they are vectors or scalars. Additionally, the explanation of GPU prefetching and pipelining is too vague, making it difficult to follow the precise implementation steps.


[1] https://hanlab.mit.edu/projects/spatten
[2] Lee, Chonghan, et al. "Token and Head Adaptive Transformers for Efficient Natural Language Processing." Proceedings of the 29th International Conference on Computational Linguistics. 2022.
[3] Ge, Suyu, et al. "Model tells you what to discard: Adaptive kv cache compression for llms." arXiv preprint arXiv:2310.01801 (2023).
[4] Zhao, Youpeng, Di Wu, and Jun Wang. "ALISA: Accelerating Large Language Model Inference via Sparsity-Aware KV Caching." arXiv preprint arXiv:2403.17312 (2024).

**Questions:**

Please answer my question in the weakness section. In addition, it would be great for the authors to provide more details in the rebuttal regarding the prefetching design.

---

### Official Review · Reviewer_VG1T · 2024-11-04

**Soundness:** 2
**Presentation:** 3
**Contribution:** 2
**Rating:** 3
**Confidence:** 4

**Summary:**

This paper introduces a new post-training sparse attention method. It implements two systems, including offloading and non-offloading ones, showing that with double-sparse, we can have high accuracy and low TPOT in different models, including MoE, GQA, and VLMs.

**Strengths:**

This paper has a solid system implementation. The comprehensive system evaluation shows that Double-Sparse can lead to over 10 times acceleration when offloading. This system has also been shown to be compatible with quantization methods and has been implemented into SGLang, a SOTA LLM serving system.

**Weaknesses:**

1 Given that SparQ is already there, the paper does not explain why the important channels are static.
2 Given that SparQ is already there, the accuracy results of SparQ are also expected to be shown, even if I can accept that SparQ is hard to have actual acceleration in GPUs.
3 InfLLM also has an on-device cache to accelerate decoding, but the paper does not well cite or discuss this fact.
4 Double-Sparse can also be seen as a special case of Loki by constraining the low-rank projectors in Loki to be a selective matrix (which means there is exactly one "1" in each row and others are zero). Given this fact, it's necessary to show the comparison with Loki, although I do not think Double-Sparse can outperform Loki.
5 As far as I know, the sequence in LongBench is not that long (256K mentioned in the paper). To make the statement fair, I will ask for accuracy evaluations on benchmarks with long enough contexts. Perperlexity is not enough, as streamingLLM can also do well.
6 The retrieval task is not standard. I will ask for evaluations on RULER to better understand the technique.

**Questions:**

As stated in Weakness.

**Details Of Ethics Concerns:**

No.

---

### Meta-Review · Area_Chair_4od8 · 2024-12-20

**Metareview:**

This paper introduces a "Double Sparsity" method for LLM efficiency by combining token and channel sparsity. While it shows promising results and practical gains, reviewers largely question its novelty and want more comparions with relevant baselines. The evaluation lacks tests on longer contexts and some popular benchmarks. Presentation also needs clarity, especially in notations and GPU details. Although one reviewer strongly supports the approach, the majority do not. Unfortuately, this leads to a rejection. In a future revision, aditional evaluations, clearer explanations, and stronger novelty claims are needed.

**Additional Comments On Reviewer Discussion:**

There is no discussion. The authors did not write a rebuttal.

---

### Decision · Program_Chairs · 2025-01-22

Reject